# Predicting black ice-related accidents with probabilistic modeling using GIS-based Monte Carlo simulation

**Seok Bum Hong**[1], **Hong Sik Yun**[1,2]*

**1** Interdisciplinary Program for Crisis, Disaster and Risk Management, Sungkyunkwan University, Suwon, Gyeonggi Province, Republic of Korea, **2** School of Civil and Architectural Engineering, Sungkyunkwan University, Suwon, Gyeonggi Province, Republic of Korea

* yoonhs@skku.edu

## Abstract

Black ice, a phenomenon that occurs abruptly owing to freezing rain, is difficult for drivers to identify because it mirrors the color of the road. Effectively managing the occurrence of unforeseen accidents caused by black ice requires predicting their probability using spatial, weather, and traffic factors and formulating appropriate countermeasures. Among these factors, weather and traffic exhibit the highest levels of uncertainty. To address these uncertainties, a study was conducted using a Monte Carlo simulation based on random values to predict the probability of black ice accidents at individual road points and analyze their trigger factors. We numerically modeled black ice accidents and visualized the simulation results in a geographical information system (GIS) by employing a sensitivity analysis, another feature of Monte Carlo simulations, to analyze the factors that trigger black ice accidents. The Monte Carlo simulation allowed us to map black ice accident occurrences at each road point on the GIS. The average black ice accident probability was found to be 0.0058, with a standard deviation of 0.001. Sensitivity analysis using Monte Carlo simulations identified wind speed, air temperature, and angle as significant triggers of black ice accidents, with sensitivities of 0.354, 0.270, and 0.203, respectively. We predicted the probability of black ice accidents per road section and analyzed the primary triggers of black ice accidents. The scientific contribution of this study lies in the development of a method beyond simple road temperature predictions for evaluating the risk of black ice occurrences and subsequent accidents. By employing Monte Carlo simulations, the probability of black ice accidents can be predicted more accurately through decoupling meteorological and traffic factors over time. The results can serve as a reference for government agencies, including road traffic authorities, to identify accident-prone spots and devise strategies focused on the primary triggers of black ice accidents.

## 1. Introduction

Black ice typically occurs in dark and cold places such as bridges, tunnel entrances, and shaded roads, and it is difficult for drivers to distinguish it from regular roads because it thinly coats

**Data Availability Statement:** All spatial files utilized in this research were sourced from public datasets available via the NGII database. The datasets were not altered or manipulated in their raw form; instead, they were reorganized into a format

suitable for the analysis conducted in this study. This reorganization process was designed to ensure the data's compatibility with our research methodologies and objectives, without compromising the integrity of the original data. The reorganized spatial files, specifically tailored for our study, are now publicly accessible under the following DOI/URL: accession numbers(DOI): 10.5281/zenodo.10863284 Link(URL) to the reorganized dataset: https://zenodo.org/records/10863284?token=eyJhbGciOiJIUzUxMiJ9.eyJpZCI6IjJkNDhlM2MzLTY0ZDQtNDhjOS04MWNlLTg5ZjUzN2UwM2NiMiIsImRhdGEiOnt9LCJyYW5kb20iOiIyZmQzNTlhOWI3ZmE0MjNlMTE5YTYwZGI0MGMxYzQxNSJ9.xbXhlQ2P_3NK0QhHdIwT0k5QH17IFvC0h-W1aOjS_-Gptbb9s-R2fIAV9oiM78GHtACnGvqrSRK4u9O0KDiXCA And can also be accessed through the supporting information files.

**Funding:** This research was also supported by the National Research Foundation of Korea (NRF) grant funded by the Korea government (MSIT) (No. 2021R1A2C201231911).

**Competing interests:** The authors have declared that no competing interests exist.

black asphalt [1]. Black ice poses risk of accidents because it forms rapidly in certain weather conditions when 'freezing rain' instantly freezes upon contact with the road and it is difficult to identify visually [2]. It has been reported that in the past five years, the number of fatalities due to black ice accidents has been four times that due to accidents caused by snowy roads, indicating that black ice-related accidents result in more deaths than accidents involving general winter road conditions [3]. The causes of black ice, which leads to traffic accidents in winter, are well understood and include fog (moisture), freezing rain, and melted snow [4, 5]. The primary cause of freezing rain is immediate freezing upon contact with the road, making it challenging to predict the occurrence, amount, and subsequent accidents. Despite the implementation of various countermeasures for accident management, traffic accidents caused by black ice have continued to occur consistently over the past five years. This implies that we are unable to respond effectively to black ice, which occurs unexpectedly and leads to accidents with a high mortality rate. Therefore, selecting road points with a high probability of black ice accidents and apply focused countermeasures is necessary. The background of this study is the southern region of Korea, which experiences temperatures dropping to below minus 10˚C in winter and has a uniform distribution of mountains and watershed areas in the inland regions.

Because of its geographical and climatic characteristics, the area frequently experiences black ice formation, which leads to accidents [3]. In January 2022, black ice led to an accident on the road towards Soheul-eup, Pocheon, on the Guri-Pocheon highway in Gyeonggi Province, in which approximately 40 vehicles were involved in a multi-car collision. This resulted in one fatality, three serious injuries, and 25 casualties [6]. In November 2021, black ice resulted in seven injuries on roads such as the Yeongdong highway in the Gangwon Province [7]. In January 2020, black ice caused a multicar collision involving 41 vehicles on a national road in Gyeongsangnam-do, resulting in ten injuries [8]. In December 2019, black ice caused a chain collision on the Sangju-Yeongcheon Expressway, resulting in 49 casualties [9]. In accidents caused by black ice, several deaths occurred, indicating a high fatality rate. Predicting the sections in which black ice accidents can occur and establishing structural or nonstructural measures to reduce this rate is essential. To implement these measures, it is critical to determine the probability of accidents caused by black ice using a geographical information system (GIS). In addition, it is necessary to formulate strategies for analyzing the factors that trigger black ice accidents based on weather, topography, and traffic conditions using accident prediction results. A sensitivity analysis of the disaster factors and damage levels is referenced in this study [10–12].

The aim of this study is to predict black ice accidents at each point through a Monte Carlo simulation on a GIS and to analyze the degree of the likelihood of an accident occurrence for each black ice incident. Unlike in previous studies, time-independent results were derived using Monte Carlo simulations based on random variables [13]. In contrast to a previous black ice study [14] that used fixed-time data for predictions, in this study, the results were derived over a broader time range owing to time invariance. In addition, accidents were statistically predicted by adding traffic factors beyond simple predictions of black ice occurrence location and amount. A numerical model was developed in the R language environment, and a Monte Carlo simulation was performed to achieve this objective. Consequently, the probability of accident occurrence was predicted and visualized using a GIS. Finally, the causes of black ice accidents were analyzed through a sensitivity analysis to establish a basis for countermeasures. A numerical model for black ice accidents was developed for Monte Carlo analysis, and the spatial and meteorological data of Suncheon and Jeollanam-do were statistically analyzed as input factors. The spatial data used are 'DEM, hill shade, road, bridge, river system, lake, angle, curvature, the IC,' the meteorological data used are 'air temperature, cloud cover, vapor pressure, precipitation, wind speed, snow (melted)' and the traffic information data used are

'velocity, traffic'. Finally, a hypothesis test (Z-test) was conducted for areas vulnerable to black ice accidents within the GIS. Using the validated model obtained through hypothesis testing, a sensitivity analysis was conducted to analyze the factors contributing to black ice accidents. Utilizing the developed numerical model for predicting the probability of black ice accidents on a road segment within the GIS and deriving the factors contributing to black ice accidents can serve as a critical resource for government agencies (e.g., road traffic authorities) to proactively identify vulnerable locations for winter accidents and establish preventive measures.

## 2. Related works

Prior studies were analyzed for differentiation before researching black ice accident modeling. Previous studies proposed low-cost sensors using electrical conductivity and GIS visualization technologies based on sensor detection for black ice detection and management [15]. This study has the capability of monitoring the black ice conditions of roads in real time based on sensors; however, because long-term predictions are not provided, there is a time limitation for devising effective countermeasures. Numerical weather prediction models have been used to predict road surfaces and traffic conditions [16]. This study provides an approximate prediction of the road surface conditions but faces limitations owing to inadequate integration with GIS and challenges in accurately calculating the number of occurrences. Previous studies created models of vehicle speed distributions suitable for various weather and traffic conditions on highways, thereby identifying more dangerous weather conditions [17]. In this study, the hazardous conditions for the occurrence of black ice were numerically predicted; however, the method had limitations in predicting the occurrences and locations of black ice based on GIS. A block diagram model has also been developed to estimate the ice index using road surface temperature, air temperature, and humidity [18–20]. These studies predicted the ice index to be within an approximate range; however, limitations existed in forecasting the occurrence and location of black ice based on GIS. These studies commonly lack methods for predicting the occurrence of black ice and the accident itself, and identifying the factors triggering black ice accidents is insufficient. Furthermore, even with predictions for the occurrence of black ice, the analysis often remains limited to the road temperature or the black ice index, and there is an issues with representing the occurrence and location of black ice in a GIS, which is linked with spatial information. Some studies have simulated the damage to buildings caused by a load of ice [21], detected the formation of black ice through sensors [22, 23], and simply estimated road temperatures [16, 24], all of which either lack applications for roads or fall short in some ways in predicting the occurrences and accidents of black ice with spatial information.

Hong et al. estimated the occurrence and location of black ice based on system dynamics and simulated the results using GIS. They provided intuitive prediction data regarding black ice events using a GIS [14]. In addition, following their study [14] estimated the number of black ice occurrences based on system dynamics, Hong et al. further refined their model by adding specific heat, latent heat, solar radiation (per hour), and snow (melted amount) as input factors to the numerical model and incorporating the concept of artificial neural networks for estimating the road temperature [13]. After analyzing prior studies, it was found that, overall, predictions for accidents caused by black ice are scarce and weather factors are limited by their dependence on time. For example, because freezing rain does not always occur at a specific time, using data from that time point in the simulation may fail to correctly estimate genuinely vulnerable sections. Therefore, there is a need for research on predicting accidents based on the characteristics of each highway section through a time-independent scenario, analyzing the factors that cause black ice accidents, and preparing primary data for countermeasure planning.

Table 1. Developments of this study compared with previous studies.

| | Study Area | Factors (added) | Scenario | Simulation | Results |
|---|---|---|---|---|---|
| **Previous Study** ① [27] | Jerash city, Jordan | - | Time-Dependent (Specific Dates) | Numerical Model | Road temperature (No GIS) |
| **Previous Study** ② [16] | Finland | - | Time-Dependent (Specific Period) | Numerical Model | Road temperature (No GIS) |
| **Previous Study** ③ [20] | Gunsan, Jeollabuk-do, South Korea | - | Time-Dependent (Specific Period) | Flow chart | Black ice index (No GIS) |
| **Previous Study** ④ [14] | Suncheon, Jeollanam-do, South Korea | - | Time-Dependent (Specific Date) | Numerical Model | Black Ice (GIS) |
| **Previous Study** ⑤ [13] | Gurye, Jeollanam-do, South Korea | - | Time-Dependent (Specific Date) | Numerical Model | Black Ice (GIS) |
| **Current Study** | Suncheon, Jeollanam-do, South Korea | Slope, Velocity, Traffic, Angle, Curvature, IC | Time-Independent (Random Distribution) | Numerical Model + Monte Carlo | Black Ice + Black Ice Accident (GIS) |

Monte Carlo simulations generate various scenarios by considering the input parameters as probability distributions independent of time and using the average of the outcomes derived from these scenarios as predictions for disaster occurrence. In relation to the Monte Carlo method applied in the field of disasters, there have been studies predicting traffic accidents using statistical input values [25, 26], and studies predicting meteorological disasters such as tornadoes and floods using statistical input values [10, 11]. These studies have provided directions for the application of Monte Carlo simulations in disaster management. However, research related to traffic accidents has not been effectively integrated with GIS and has not primarily focused on black ice accident cases. Considering prior research, it was deemed necessary to apply Monte Carlo simulations to the field of black ice accident prediction, particularly by utilizing meteorological and traffic factors as key inputs. Previous research has raised the question of which theories should be used to predict and respond to traffic accidents caused by black ice. This emphasizes the need for a GIS-based analysis of black ice occurrences and locations, numerical modeling for black ice accident probability, and the use of Monte Carlo simulations for time independence.

Table 1 lists the improvements of this study compared with recent research. As shown in Table 1, the scientific contribution of this study has several respects: the introduction of a GIS in the field of road temperature prediction, the enhancement of the simple black ice index by predicting the occurrences and locations of black ice, the prediction of the probability of black ice accidents beyond mere forecasts of black ice occurrences, and the conduction of time-independent simulations (Monte Carlo simulation) by inputting traffic and meteorological factors with probabilistic distributions owing to their uncertainty.

## 3. Methodology

To predict black ice accidents within the GIS, a Monte Carlo simulation was conducted following the flow outlined in Fig 1 to analyze measures related to the factors contributing to black ice accidents. The Monte Carlo simulation algorithm is based on Repeated Random Sampling, which mathematically approximates the values of the designed function's values [12, 28–30]. Monte Carlo simulation is a technique that allows for the input of statistical data into factors (air temperature, cloud cover, vapor pressure, precipitation, wind speed, snow, velocity, and traffic) harboring uncertainties in the numerical model, thereby facilitating the trial of various scenarios. Once the black ice accident probabilities for different scenarios are calculated, the mean can be computed to derive the final prediction value. Therefore, in this study, we developed a model (F(x)) to incorporate random variables, performed Monte Carlo simulations to

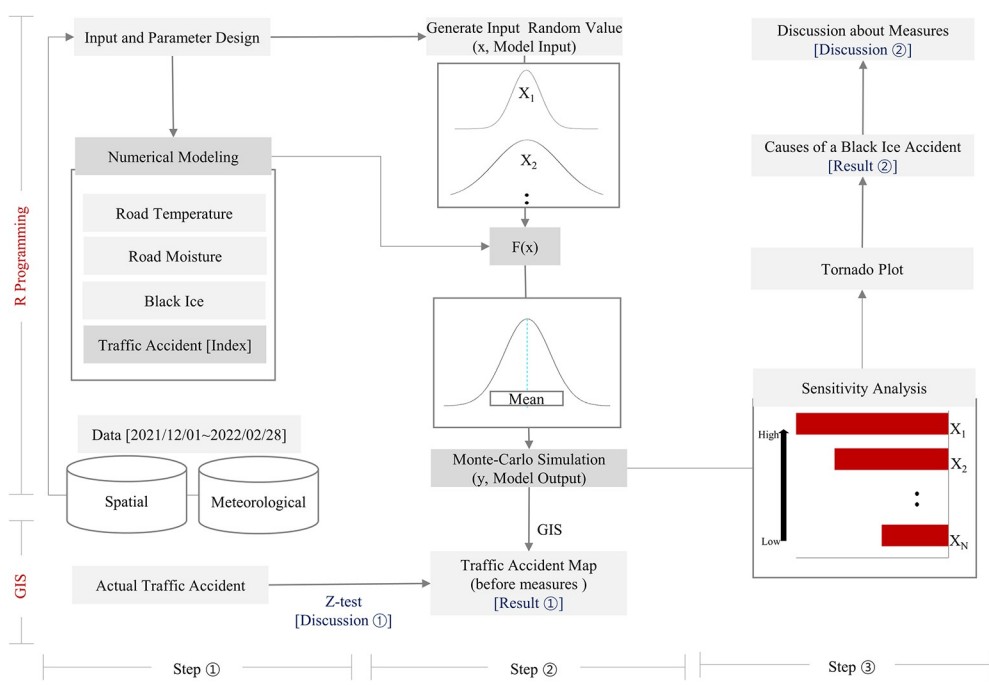

**Fig 1. Overall research flow.**

predict the occurrence probability of black ice accidents, and analyzed the contributing factors. This study consisted of three steps. Step ① involves the numerical modeling of black ice accidents. Step ② involves Monte Carlo simulations based on the numerical model (F(x)) and the validation of the model. Step ③ involves analyzing the factors contributing to black ice accidents in the model to provide a basis for establishing countermeasures. The methodologies for each step are detailed in Sections 2.1–2.3. In the research flow shown in Fig 1, three results and two corresponding discussion processes are derived. The results include a black ice accident map before the introduction of countermeasures (Result ①) and an analysis of the factors contributing to black ice accidents (Result ①), as explained in Section 3. The discussion includes the hypothesis testing of the numerical model for actual accident-prone areas (Discussion ①) and the discussion of factors contributing to black ice accidents (Discussion ②), which are explained in Section 4.

## 3.1. Numerical model for predicting black ice accidents

A numerical model was developed as a preliminary step for predicting the probability of black ice accidents using a Monte Carlo simulation within a GIS. The algorithm input and Monte Carlo simulation for numerical modeling were conducted using R Studio software, an R language development environment. A numerical model was designed to determine the black ice accident probability (BA) as a factor related to black ice accidents. The systemic causal relationships between these factors are shown in Fig 2. Fig 2 shows a causal map depicting the structure of the numerical model, where the +/- signs within the causal map indicate causal or proportional relationships between the factors [31]. The black ice accident probability, calculated based on the causal map, was visualized in the GIS as a black ice accident map divided into road segments (16,415). The calculated black ice accident probability was visualized in GIS. The purpose of the visualization in GIS is to display the computed black ice accident

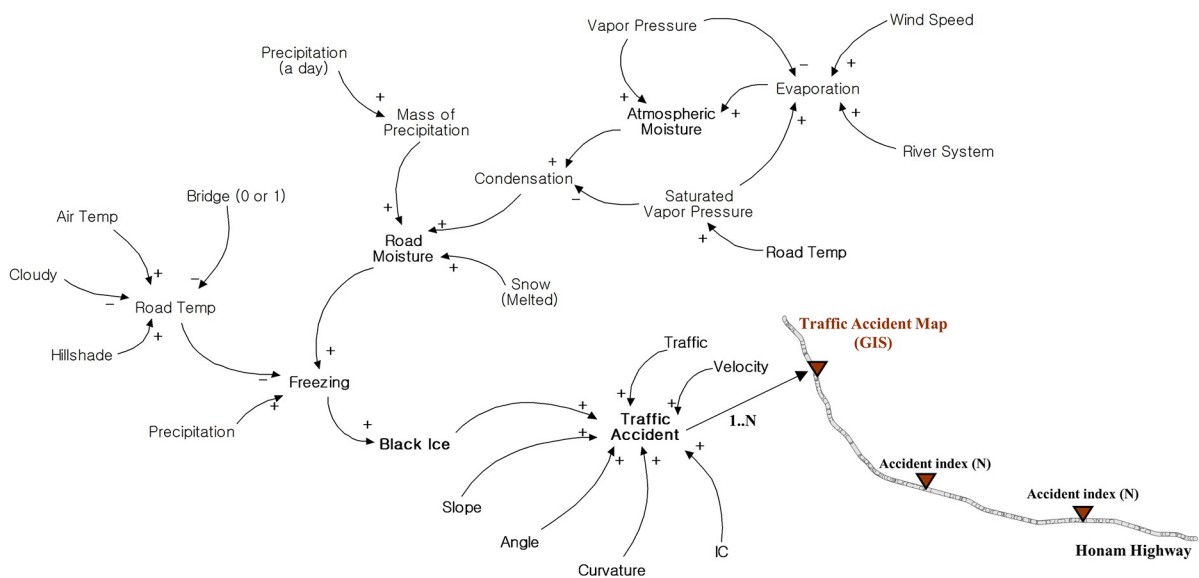

**Fig 2. Systemic causal relationships in the black ice accident model (F(x)).**

probability by location, thereby aiding in the formulation of countermeasures for specific sections. A description of each factor in the numerical model is presented in Table 2. Regarding the spatial input, the actual data for each road segment in the study area were input into the model. Regarding the meteorological and traffic information inputs, random variables (following Gaussian or Poisson distributions) were generated based on the statistical values (mean and standard deviation) of the actual data for the entire study area and then input into the model. The calculated intermediate factors include evaporation, condensation, road temperature, road moisture, and black ice, whereas the final output is BA, which represents the probability of black ice accidents and is expressed in the range of 0–1. As shown in Table 2, the spatial input was preprocessed in GIS and then entered into the system dynamics. The meteorological and traffic information inputs were directly entered into the system dynamics as statistical data. The final output data, that is, the black ice accident probability, were calculated using system dynamics and visualized using GIS.

The output factor BA represents the probability of black ice accidents and serves as the final result of the function F(x). Function F(x) is constructed according to Eq (1). F(x) is a function of the variables black ice (BI), velocity (V), traffic (T), slope (SL), angle (A), curvature (CV), and IC, with each variable having been normalized before being substituted into the equation. The symbols for each variable are denoted by $BI_N$, $V_N$, $T_N$, $SL_N$, $A_N$, $CV_N$, and $IC_N$, respectively. The maximum value for the black ice accident probability (F(x)) is 1, and the value of $BI_N$ acts as the main weight in determining the accident occurrence probability. $SL_N$ and $CV_N$ represent the probabilities influenced by the terrain environment, whereas $A_N$ and $IC_N$ represent the probabilities influenced by changes in the direction. Sli represents the probability affected by the distance to the leading vehicle, assuming that the braking distance can be increased by up to nine times when black ice occurs. Eq (2) was established to calculate this value, i.e., if the intervehicle space (1000/(T/V)) was less than nine times the vehicle speed, it was assigned a value of 1; otherwise, it was assigned a value of 0 [32]. The final calculated output factor, BA, represents the probability of accidents and is visualized in GIS as the black ice accident map segmented by road sections. The calculated results of the variables are expressed

**Table 2. Key input and output factors of the black ice accident model (F(x)).**

| Factor | Category | Symbol | Value Range | Unit |
|---|---|---|---|---|
| Hill shade (from DEM, 5 m) | Spatial Input (Preprocessing with GIS for System Dynamics Input) | HS | 0–254 | - |
| Bridge | | B | 0–1 | - |
| River System | | RS | 0–1 | - |
| Lake | | L | 0–1 | - |
| Slope | | SL | 0–90 | degree |
| Angle | | A | 0–90 | degree |
| Curvature | | CV | -0.5–0.5 | - |
| IC | | IC | 0–1 | - |
| Air Temp | Meteorological Input (System Dynamics Input) | AT | -15–10 | ˚C |
| Cloud Cover | | CL | 0–10 | - |
| Vapor Pressure | | VP | 0– | Pa |
| Precipitation | | P | 0– | mm |
| Wind Speed | | WS | 0– | m/s |
| Snow (Melted) | | SN | 0– | cm |
| Velocity | Traffic information Input (System Dynamics Input) | V | 0– | m/s |
| Traffic | | T | 0– | v/h |
| Evaporation | Intermediate (System Dynamics Calculation) | E | 0– | g/m$^2$ |
| Condensation | | CD | 0– | g/m$^2$ |
| Road Temperature | | RT | 0– | ˚C |
| Road Moisture | | RM | 0– | g/m$^2$ |
| Black Ice | | BI | 0– | g/m$^2$ |
| Black Ice Accident Probability | Output (GIS Visualization) | BA | 0–1 | - |

as a sequence (n) of 16,415 points.

$$\{BA\}_n = \{F(x)\}_n = \{F(BI, V, T, SL, A, CV, IC)\}_n$$
$$= \left\{BI_N \frac{0.5(SL_N + CV_N) + 0.5(A_N + IC_N) + Sli}{3}\right\}_n, (1 \leq n \leq 16,415) \quad (1)$$

$$Sli = \begin{cases} 1, \dfrac{1000}{\frac{T}{V}} < 9V \\[2ex] 0, \dfrac{1000}{\frac{T}{V}} \geq 9V \end{cases} \quad (2)$$

The variables V, T, and SL were calculated using spatial analysis functionalities in GIS and input into R Studio to estimate F(x). The BI was calculated by inputting the relevant spatial and weather data into the numerical model. The BI calculation is described in Eq (3). In Eq (2), BI is a function of the road temperature (RT), road moisture (RM), and precipitation (P). When P = 0, freezing occurs if the RT is below 1˚C. When P exceeds 0, freezing occurs when RT is below 0˚C [23]. The term 'freezing,' which is synonymous with BI, was determined by

multiplying the ratio of the current value of RT by the maximum value of RT.

$$\{BI\}_n = \{F(RT, RM, P)\}_n = \begin{cases} \dfrac{RT-1}{-11}RM, \text{when } P > 0 \text{ if } RT \leq 1 \\ \dfrac{RT-1}{-11}RM, \text{when } P = 0 \text{ if } RT \leq 0 \\ 0, else \end{cases}_n, (1 \leq n \leq 16,415) \quad (3)$$

The factor RT, which is one of the components of BI in Eq (3), was calculated according to Eq (4). In Eq (4), RT is a function of the air temperature (AT), cloud cover (CL), hill shade (HS), and bridge (B). Each corresponding factor, denoted as $AT_N$, $CL_N$, $HS_N$, and $BL_N$, was normalized and input into the function. The parameters W1, W2, W3, and b were multiplied by each input factor (AT, CL, and HS) and determined based on Hong et al. [13]. In this study, we established a regression equation for the road temperature using the air temperature, cloud cover, and hill shade as input factors. The parameters were trained using an evolutionary algorithm. The resulting values for W1, W2, W3, and b were calculated as 20.22, 0.18, 5.64, and -10.73, respectively. W4, representing the weight of the bridges, was set to 2 based on a literature review, which indicated that bridge temperatures are typically approximately 2°C lower than the road temperature [33]. A regression equation incorporating the aforementioned parameters was used to predict the road temperature, which was then compared with the actual values. The resulting Root Mean Square Error (RMSE) was 1.24, and the Mean Absolute Percentage Error (MAPE) was 0.07. Based on the parameters validated in previous studies, an RT function was formulated for the Monte Carlo simulations in this study.

$$\{RT\}_n = \{F(AT, CL, HS, BL)\}_n = \{W_1 AT_N + W_2 CL_N + W_3 HS_N - W_4 B + b\}_n, (1 \leq n \leq 16,415) \quad (4)$$

In Eq (3), the second input factor for calculating BI is RM, which is determined using Eq (5). RM is a function of condensation (CD), precipitation (P), and snow (SN). P (mm) was converted to mass $(g/m^2)$ and input into the model, denoted as MoP. SN was multiplied by 920 to convert SN (cm) into mass $(g/m^2)$ per unit area.

$$\{RM\}_n = \{F(CD, P, SN)\}_n = \{CD + MoP + 920SN\}_n, (1 \leq n \leq 16,415) \quad (5)$$

In Eq (5), the precipitation-related factors are the base factor precipitation (P) and the calculated mass of precipitation (MoP). MoP in Eq (5) represents the actual precipitation mass per unit area per hour in a volume of 1 $m^3$. This is further explained in Eq (6), where P represents the precipitation in millimeters (mm), which is converted into centimeters (cm) when calculating the mass.

$$MoP(g/m^3) = \frac{\pi r^2 P}{10} \quad (6)$$

The condensation (CD) in Eq (5) was calculated based on the relationship between the current amount of water vapor in the atmosphere and the saturated water vapor content on the road. Condensation occurs when the temperature of the water vapor in the atmosphere falls below the dew point of the road surface. Eq (7) is the condensation equation, where E represents the amount of evaporation $(g/m^3)$, AoV represents the amount of water vapor $(g/m^3)$, and AoSV represents the saturated water vapor content $(g/m^3)$.

$$C = \sum_{t=0}^{t=13}(E + AoV - AoSV), \quad (7)$$

The evaporation term in Eq (7) is based on Dalton's law, which is a fundamental aerodynamic

principle that explains evaporation [34]. This theory states that the movement of water molecules on a free surface is proportional to the vertical vapor pressure gradient. This is expressed by Eq (8), where E represents the amount of evaporation from the reservoir, $e_s$ is the saturation vapor pressure at air temperature (mmHg), $P_v$ is the actual vapor pressure at air temperature (mmHg), and Sw is the wind speed 2 m above the water surface (m/s). The vapor amount (AoV) in Eq (7) was obtained using Eq (9), where VP represents the vapor pressure and AT represents the air temperature.

$$E = 0.345(e_s - P_v)(0.5 + 0.54S_w) \tag{8}$$

$$AoV = 217\frac{VP}{AT + 273.15} \tag{9}$$

### 3.2. Prediction of black ice accidents

In step ①, a numerical model was developed to predict the black ice accident probability. In step ②, scenarios were selected, and random variables were generated based on them for Monte Carlo simulations in the numerical model. In the scenario selection and random variable generation sections, probability distributions were established using Gaussian and Poisson distributions, and random variables were extracted accordingly. In the field of black ice accident prediction based on Monte Carlo simulations, the extracted random variables are input into the developed model to derive the occurrence probability distribution for black ice accidents and calculate an average value. The extracted BA values were visualized using GIS, and their utility was validated through hypothesis testing (Z-test).

**3.2.1. Scenario-based data and random variable construction.** Before conducting the Monte Carlo simulations, Suncheon, Jeollanam-do, South Korea was selected as the study area for the statistical analysis and simulations. Suncheon, located adjacent to the coast, is known for its high rainfall due to its geographical conditions. The annual precipitation in Suncheon is 1,308 mm, which is among the highest in Jeollanam-do Province. Additionally, numerous water bodies (e.g., lakes and rivers) contribute to high humidity and frequent fog formation. The inland areas of Suncheon are characterized by evenly distributed mountains, with the highest peak reaching approximately 942 m. Given the combination of mountainous regions that create hill shade and water bodies that contribute to high humidity, it was anticipated that black ice formation would frequently occur in Suncheon. Fig 3 illustrates the location of the Honam Expressway in Suncheon, which was selected as part of the study area. The Honam Expressway in Suncheon is generally surrounded by mountains on both sides ((a)-(b)) and is adjacent to water bodies. Some sections of the Honam Expressway in Suncheon are approximately 24 km long, spanning points A (127°14') to B (127°29'). The spatial variables include 'hill shade, bridge, river system, lake, slope, angle, curvature, IC' as part of the spatial input. The hill shade, slope, and curvature were derived from DEM data with a resolution of 5 m. The statistical values of the key spatial variables are listed in Table 3.

The DEM for the hill shade from among the spatial variables in Table 3 was acquired through lidar satellites of Korea's National Geographic Information Institute [35]. The hill shade was generated based on the DEM using ArcGIS Pro software. Additionally, the remaining spatial variables, such as bridges, river systems, lakes, slopes, angles, curvatures, and IC, were obtained from Korea's National Spatial Data Infrastructure Platform (NSDIP) [36]. All spatial variables were organized using ArcGIS Pro software and then input into R Studio for the simulation.

After selecting the scenarios, the actual data of the winter season in Suncheon, Jeollanam-do, from December 1, 2021, to February 28, 2022, were acquired to extract random variables

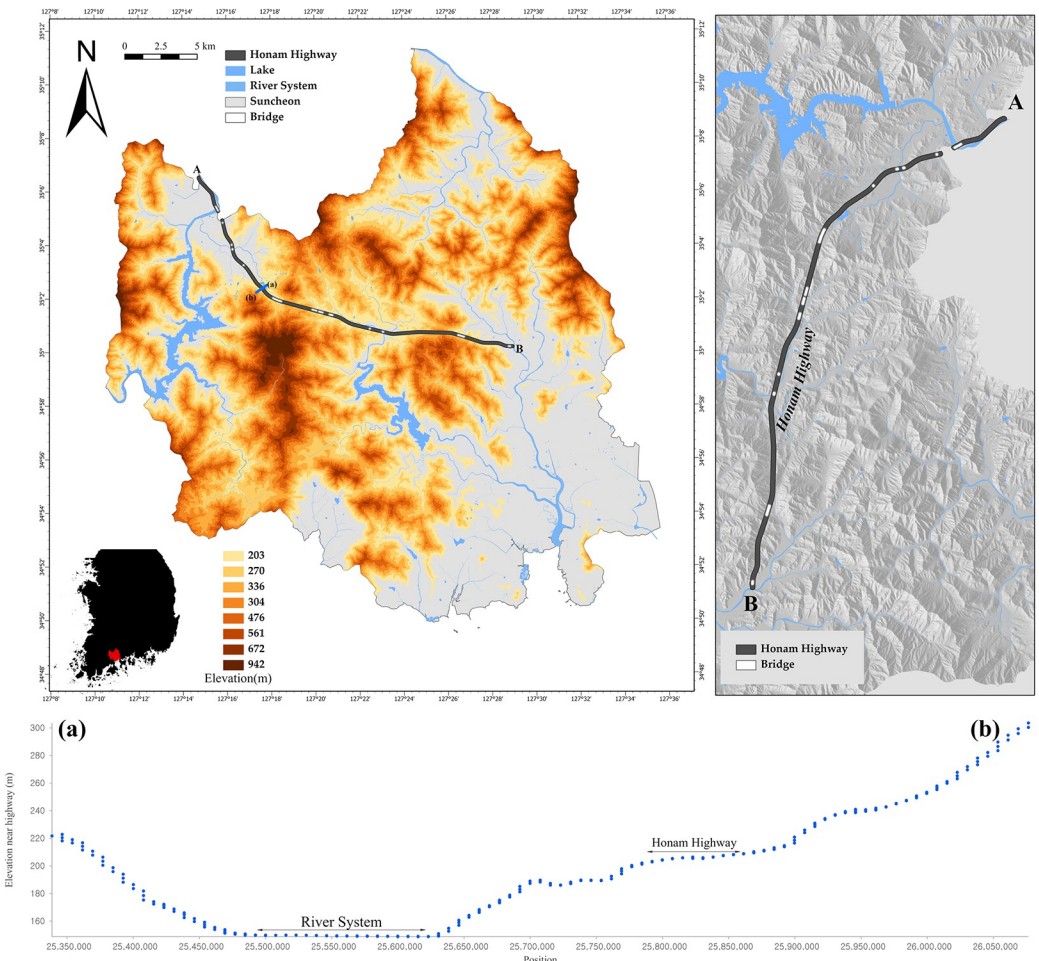

**Fig 3. Honam highway and spatial information of Suncheon, Jeollanam-do.** (Source: DEM created by the LIDAR method at the National Geographic Information Institute in Korea, https://www.ngii.go.kr/kor/content.do?sq=204).

using Gaussian and Poisson distributions. The factors included air temperature, cloudy weather, vapor pressure, precipitation, wind speed, snow (melted), velocity, and traffic; 1000 random variables were extracted for each factor. Traffic was constructed as a random variable using a Poisson distribution, representing the number of specific events occurring within a unit time interval. The Poisson distribution is described by Eq (10). Eq (10) presents the

**Table 3. Statistics of spatial inputs.**

| Factor | Resolution | Reference | Mean (μ) | Standard Deviation (σ) | Unit |
|---|---|---|---|---|---|
| Hill shade (From DEM, 5 m) | 5m | Lidar Satellite | 86 | 19.77 | - |
| Bridge | 1m | NSDIP | 0.0092 | 0.074 | - |
| River System | 1m | NSDIP | 0.22 | 0.27 | - |
| Lake | 1m | NSDIP | 0.79 | 0.27 | - |
| Slope | 1m | NSDIP | 6.07 | 4.39 | ° |
| Angle | 1m | NSDIP | 1.92 | 1.67 | ° |
| Curvature | 1m | NSDIP | 0.50 | 2.37 | - |
| IC | 1m | NSDIP | 0.02 | 0.15 | - |

**Table 4. Statistics of meteorological and traffic inputs.**

| Factor | Distribution | Period | Mean (μ) | Standard Deviation (σ) | Unit |
|---|---|---|---|---|---|
| Air Temperature | Gaussian | 2021/12/01~2022/02/28 | 1.16 | 4.97 | ˚C |
| Cloud Cover | Gaussian | 2021/12/01~2022/02/28 | 4.3 | 4.01 | - |
| Vapor Pressure | Gaussian | 2021/12/01~2022/02/28 | 4.12 | 1.66 | Pa |
| Precipitation | Gaussian | 2021/12/01~2022/02/28 | 0.53 | 0.5 | mm |
| Wind Speed | Gaussian | 2021/12/01~2022/02/28 | 2.79 | 1.88 | m/s |
| Snow | Gaussian | 2021/12/01~2022/02/28 | 0.23 | 0.13 | cm |
| Velocity | Gaussian | 2021/12/01~2022/02/28 | 94 | 7.65 | m/s |
| Traffic | Poisson | 2021/12/01~2022/02/28 | 764 | 1020.2 | v/h |

probability distribution of the number of events occurring K times within a fixed period, given an expected value λ for the number of events occurring [37]. For the Poisson distribution of traffic, the average (μ) value of the actual data was substituted into the parameter. The remaining factors correspond to continuous variables and are constructed as random variables using a Gaussian distribution, commonly known as a normal distribution [38]. In a Gaussian distribution, the mean, mode, and median are all μ, and the expected value of the normal distribution is given by Eq (11). The conditions for the probability distributions constructed for each factor are listed in Table 4. A normal distribution was used to approximate the distribution of the collected continuous data by relying on the central limit theorem, which states that the means of random variables tend to follow a normal distribution [39]. In this study, random variables were generated for the relevant continuous factors. Table 4 presents the mean (μ) and standard deviation (σ) for each factor. The Meteorological and Traffic factors listed in Table 4 were acquired through Korea's "KMA Weather Data Service—Open MET Data Portal" and the "Expressway Public Data Portal" [40, 41]. The Meteorological and Traffic data were obtained in text format and then input into R studio, where the Monte Carlo simulation was performed. The mean (μ) and standard deviation (σ) were calculated for the data acquisition period (2021/12/01~2022/02/28) to organize the data for the input into the numerical model in R studio. All these datapoints can be treated as contextual information, and their analysis can inform the formulation of future strategies.

$$f(k; \lambda) = \frac{\lambda^k e^{-\lambda}}{k!} \qquad (10)$$

$$\bar{X} = \int_{-\infty}^{\infty} \frac{x}{\sigma\sqrt{2\pi}} \exp\left[-\frac{(x-\mu)^2}{2\sigma^2}\right] \qquad (11)$$

The distributions of the random variables generated based on the mean (μ) and standard deviation (μ) in Table 4 are shown in Fig 4. Variables (a)–(g) are continuous variables constructed using a Gaussian distribution to generate random values, and variable (h) is built using a Poisson distribution. Except for the air temperature, the other variables ((b) to (h)) have a range of values greater than or equal to zero. Therefore, any negative values were converted to zero when input into the model.

**3.2.2. Monte Carlo simulation-based accident prediction and validation.** Scenarios and random variables were constructed, and Monte Carlo simulations were performed based on the numerical model. The Monte Carlo simulation is a mathematical technique used to predict the possible outcomes of uncertain events. The occurrence of black ice and the probability of

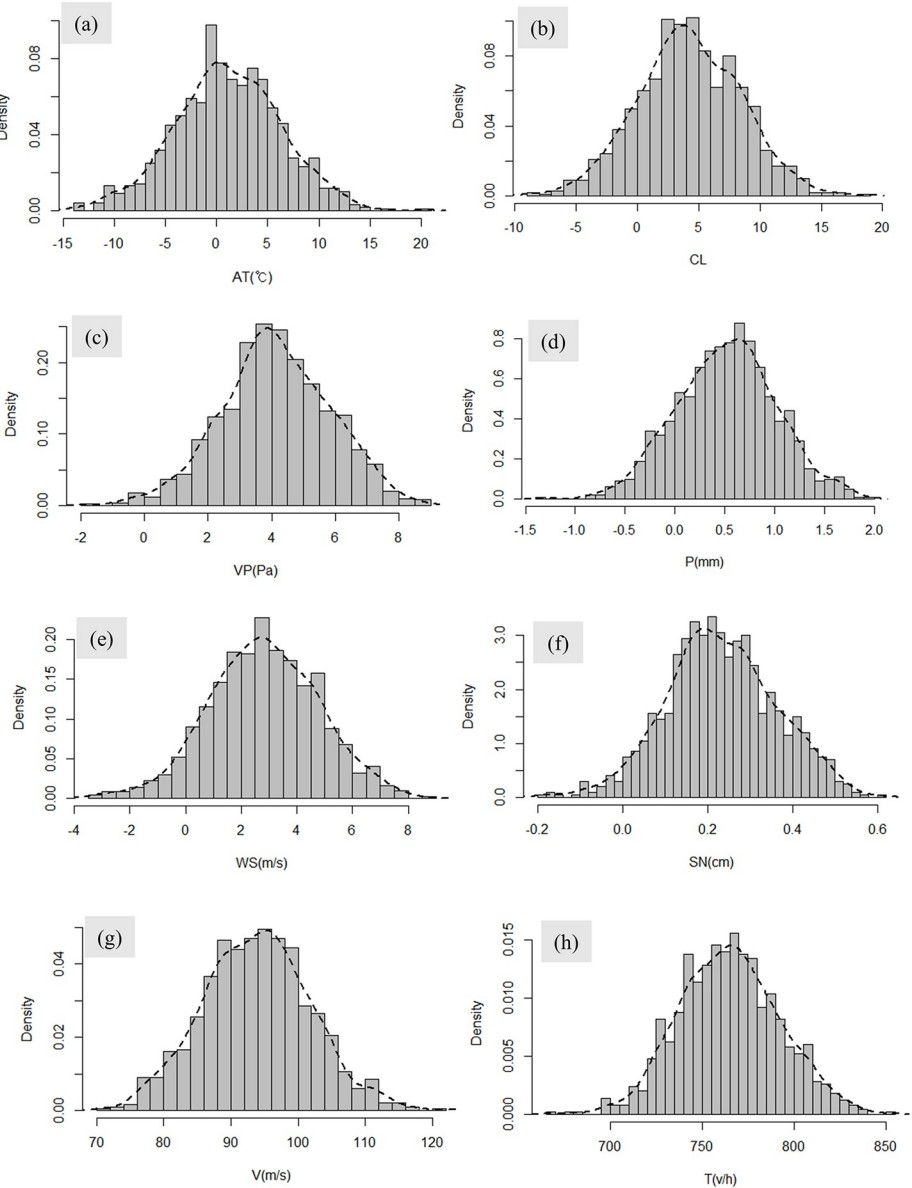

**Fig 4.** Construction of random values for black ice factors: (a) Gaussian distribution of air temperature, (b) Gaussian distribution of cloud cover, (c) Gaussian distribution of vapor pressure, (d) Gaussian distribution of precipitation, (e) Gaussian distribution of wind speed, (f) Gaussian distribution of snow (melted), (g) Gaussian distribution of velocity, and (h) Poisson distribution of traffic.

accidents vary depending on the values of the variables, making their numerical values uncertain. Therefore, a statistical analysis of the variable range is necessary according to the Monte Carlo simulation methodology to determine the numerical range for black ice occurrence and accident probability. The numerical range for the accident probability was calculated and visualized on a road-segment basis in a GIS with 16,415 classified road segments, and Monte Carlo simulations were conducted for each segment. The prediction of black ice accidents based on the Monte Carlo simulation was conducted using the numerical model represented by F(x) in Step ② of Fig 1. Randomly generated values of the input factors were fed into the model, and a probabilistic distribution of black ice accidents was obtained as output. The resulting black ice

accident probability was visualized in the GIS at the road point level, corresponding to the black ice accident map shown in Fig 2 (Result ①). The Monte Carlo analysis serves two purposes. First, it provides predictions of black ice accidents by road segment (Result ①), as described previously. Second, it analyzes the factors contributing to black ice accidents through a sensitivity analysis (Result ②). The identified factors include 'hill shade, slope, angle, curvature, IC, air temp, cloud cover, vapor pressure, precipitation, wind speed, snow (melted), velocity, and traffic,' which are input into the numerical model F(x) as X1~X8.

## 4. Results

Following the research methodology described above, the following two results were obtained: Result ① represents the accident map before the implementation of any countermeasures. Result ② includes sensitivity values and a list of influential factors extracted based on the Monte Carlo simulation's sensitivity analysis. The scientific or societal contribution of Result ① lies in its ability to identify specific sections of expressways that are highly vulnerable to black ice accidents, thereby aiding in the implementation of countermeasures. Additionally, Result ② offers a scientific contribution by enabling a detailed exploration of specifics when implementing countermeasures.

### 4.1. Black ice accident map (Result ①)

Based on the Monte Carlo simulation results, an accident map was constructed before the implementation of countermeasures. The resulting graphs of the Monte Carlo simulations are shown in Fig 5. Fig 5(A) shows the probability distribution of black ice occurrence at 16,415 road locations. The average value for black ice was 59.68 $g/m^2$, with a standard deviation of 15.25 $g/m^2$. This average value corresponds to the approximation of black ice predicted using the Monte Carlo simulation. Similar to Fig 5(A) and 5(B) shows the probability distribution of the black ice transient accident occurrence at 16,415 road locations. The mean value was 0.0058, with a standard deviation of 0.0019. This mean value corresponds to the predicted approximation of the occurrence probability obtained from the Monte Carlo simulations.

The black ice accident probability results derived from the Monte Carlo simulation were visualized on a GIS platform, as shown in Fig 6. The black ice accident probabilities were mapped with an average value of 0.0057 and a standard deviation of 0.0019. These values range from 0 to 1 and are relative. Fig 7 shows a map that highlights the results of using the buffer function to increase the width of the existing roads by 45 m on both sides, consisting of 16,415 points.

### 4.2. Sensitivity analysis of black ice accident factors (Result ②)

The sensitivity analysis results, based on the Monte Carlo simulation, are presented in Table 5, which presents a magnitude between 10% and 90% for the black ice accident probability (BA) values determined by the 12 key factors and the base sensitivity calculations for these values. In descending order, the three factors with the highest sensitivity were wind speed, air temperature, and angle. The factor with the highest weight was wind speed, with a sensitivity of 0.354. This analysis indicates the range of values associated with this factor and the resulting changes to the black ice accident probability. In other words, a higher sensitivity was observed when there was a larger variation in the factor values and a greater corresponding change in the black ice accident probability. The analysis of factors with a high sensitivity (such as wind speed, air temperature, and angle) was deemed to be important and valuable. This is because the results can be used to select focus areas for the efficient allocation of budgets for countermeasures.

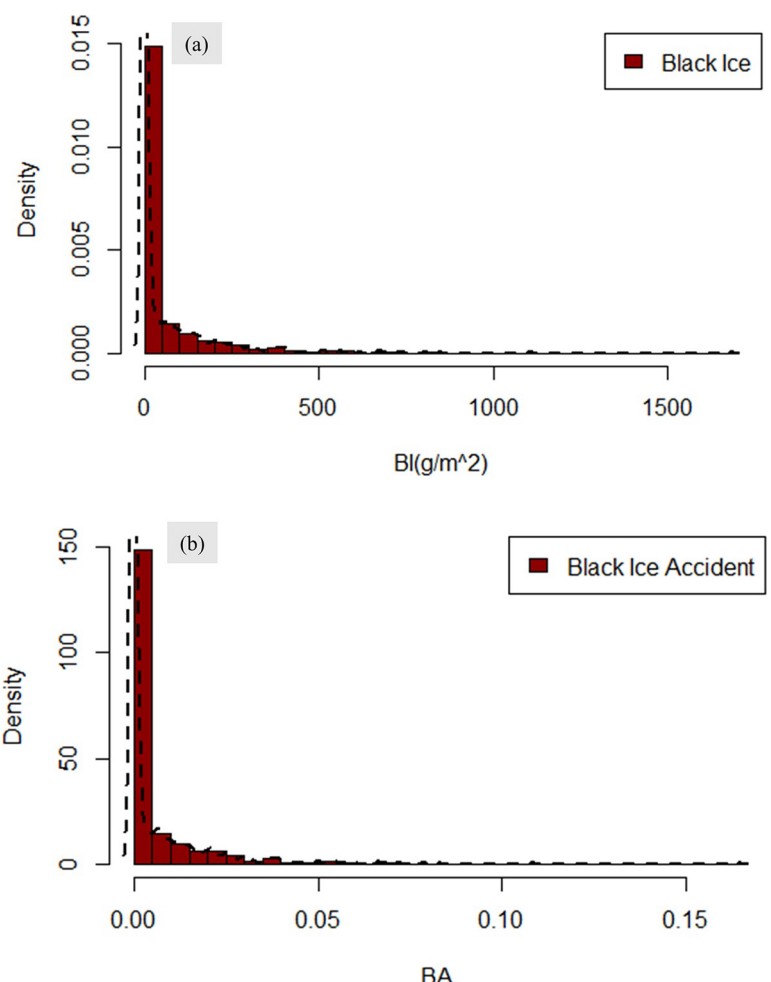

**Fig 5.** Graph of estimated black ice accident probability: (a) Histogram of 'Black ice,' and (b) Histogram of 'Black ice accident probability'.

## 5. Discussion

In this study, two key results were obtained: a black ice accident map and a sensitivity analysis of black ice accident factors. The first result, a black ice accident map, identified vulnerable sections with a high probability of black ice accidents using GIS. The average probability of black ice accidents was found to be 0.0058, with a standard deviation of 0.0019. The second result, 'Sensitivity Analysis of Black Ice Accident Factors,' revealed that 'wind speed, air temperature, and angle' had respective sensitivities of 0.085, 0.200, and 0.270. The scientific question addressed in this study is which theory is most effective for accurately predicting and responding to traffic accidents caused by black ice. The scientific and societal contributions of this study derive from enabling the identification of vulnerable points for black ice accidents based on Monte Carlo simulations and by facilitating the exploration of specific details for countermeasures through sensitivity analysis. To validate the results, the numerical model for black ice accident probability is verified in Section 5.1, and the countermeasures for each factor are analyzed in Section 5.2 according to their sensitivity ranking. Prior to the detailed discussion, Table 6 presents the main lessons and areas for development as identified from the conclusions. Both results were limited by simulations within fixed-time scenarios, indicating the

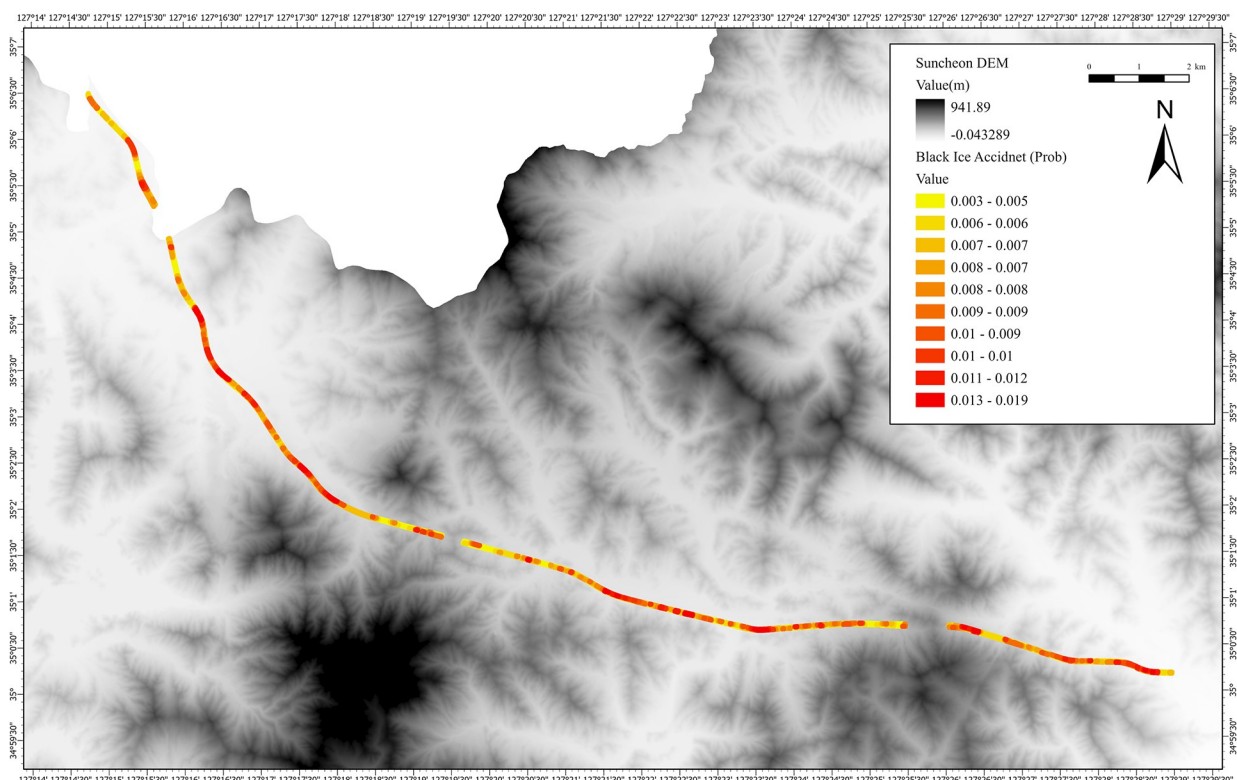

**Fig 6.** Estimation map of black ice accident probability. (Source: DEM created by the LIDAR method at the National Geographic Information Institute in Korea, https://www.ngii.go.kr/kor/content.do?sq=204).

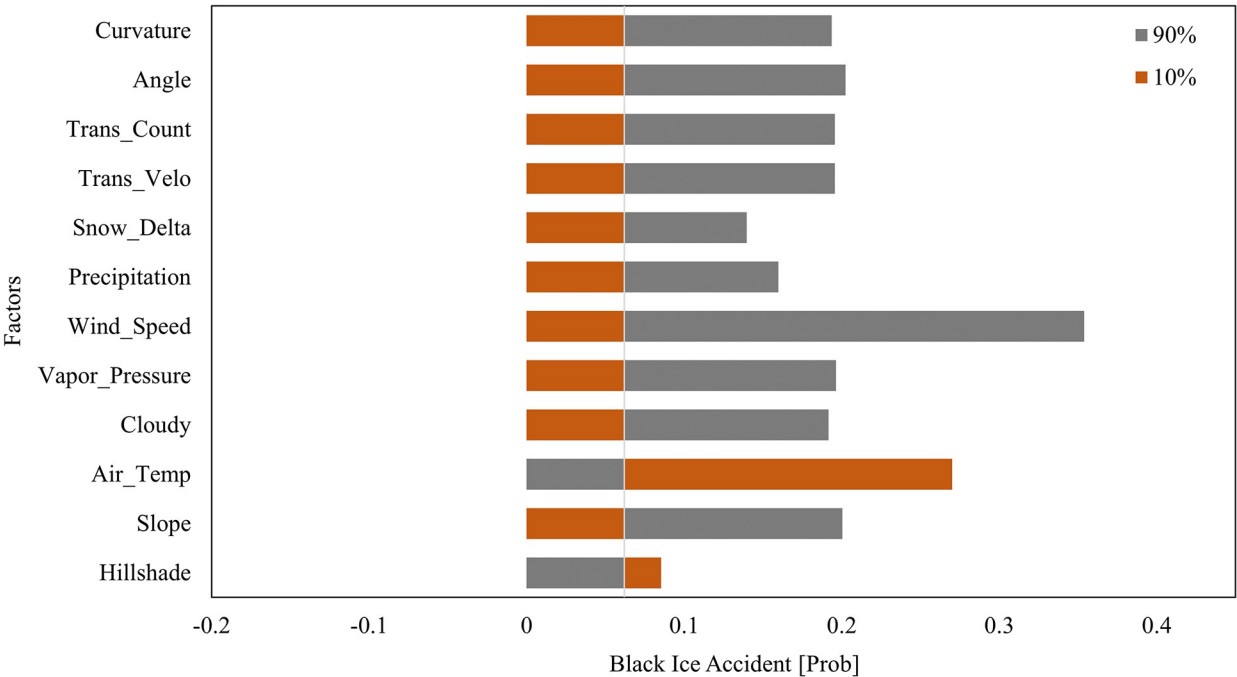

**Fig 7. Tornado plot of black ice accident sensitivity analysis.**

Table 5. Sensitivity analysis results.

| Factors | 10% | Median | 90% | Sensitivity | Rank |
|---|---|---|---|---|---|
| Hill Shade | 0.085 | 0.69 | 0 | 0.085 | 12 |
| Slope | 0 | 0.69 | 0.200 | 0.200 | 4 |
| Air Temperature | 0.270 | 0.69 | 0 | 0.270 | 2 |
| Cloud Cover | 0 | 0.69 | 0.192 | 0.192 | 9 |
| Vapor Pressure | 0 | 0.69 | 0.196 | 0.196 | 5 |
| Wind Speed | 0 | 0.69 | 0.354 | 0.354 | 1 |
| Precipitation | 0 | 0.69 | 0.160 | 0.160 | 10 |
| Snow (Melted) | 0 | 0.69 | 0.140 | 0.140 | 11 |
| Velocity | 0 | 0.69 | 0.196 | 0.196 | 6 |
| Traffic | 0 | 0.69 | 0.196 | 0.196 | 6 |
| Angle | 0 | 0.69 | 0.203 | 0.203 | 3 |
| Curvature | 0 | 0.69 | 0.194 | 0.198 | 8 |

need for future research to conduct time-series predictions of black ice accidents and perform sensitivity analysis under time-series conditions.

## 5.1. Hypothesis testing of the numerical model for actual accident-prone areas (Discussion ①)

To validate the black ice accident probability estimation model based on the Monte Carlo simulation, we selected the Jeobchijae 1, Jeobchijae 2, and Ssangamjae sections of the Honam Expressway and performed hypothesis testing using actual accident statistics. The Jeobchijae 1 and Jeobchijae 2 sections and Ssangamjae are segments of the Honam Expressway located in Suncheon that have recorded a high frequency of black ice accidents. The results are presented in Fig 8. Fig 8(A) shows the black ice accident map of the Jeobchijae 1 section with a mean of 0.0065 and a standard deviation of 0.0018. Fig 8(B) shows the black ice accident map of the Jeobchijae 2 section with a mean of 0.0065 and a standard deviation of 0.0014. Fig 8(C) shows the black ice accident map of Ssangamjae with a mean of 0.0058 and a standard deviation of 0.0017.

A Z-test was employed to validate the significance of the black ice accident probability values of the Jeobchijae 1, Jeobchijae 2, and Ssangamjae sections. The Z-test is a statistical analysis technique that compares the means of a population and sample group to test a hypothesis [42]. The formula for standardizing the observed value X (black ice accident probability) to calculate the Z-score is given by Eq (12) and Ref. [43], where X represents the variable for the black ice accident probability, μ denotes the population mean of the black ice accident probability, σ represents the population standard deviation, and n is the number of variables. In this study, a Z-test was employed to verify the difference in accident rates between high-frequency and general zones. For hypothesis testing, we formulate H0 and H1 as follows:

$$Z = \frac{X - \mu}{\frac{\sigma}{\sqrt{n}}} \tag{12}$$

Table 6. Summary of research findings and future directions.

| | Title | Type | Contribution | Limitations | Future Research |
|---|---|---|---|---|---|
| Result ① | Black Ice Accident Map | GIS map | Vulnerability Classification | Fixed Time | Time-Series Prediction of Black Ice Accidents |
| Result ② | Sensitivity of Black Ice Accident Factors | Tornado plot | Efficiency Improvement in Countermeasure Development | Fixed Time | Sensitivity Analysis in Time Series |

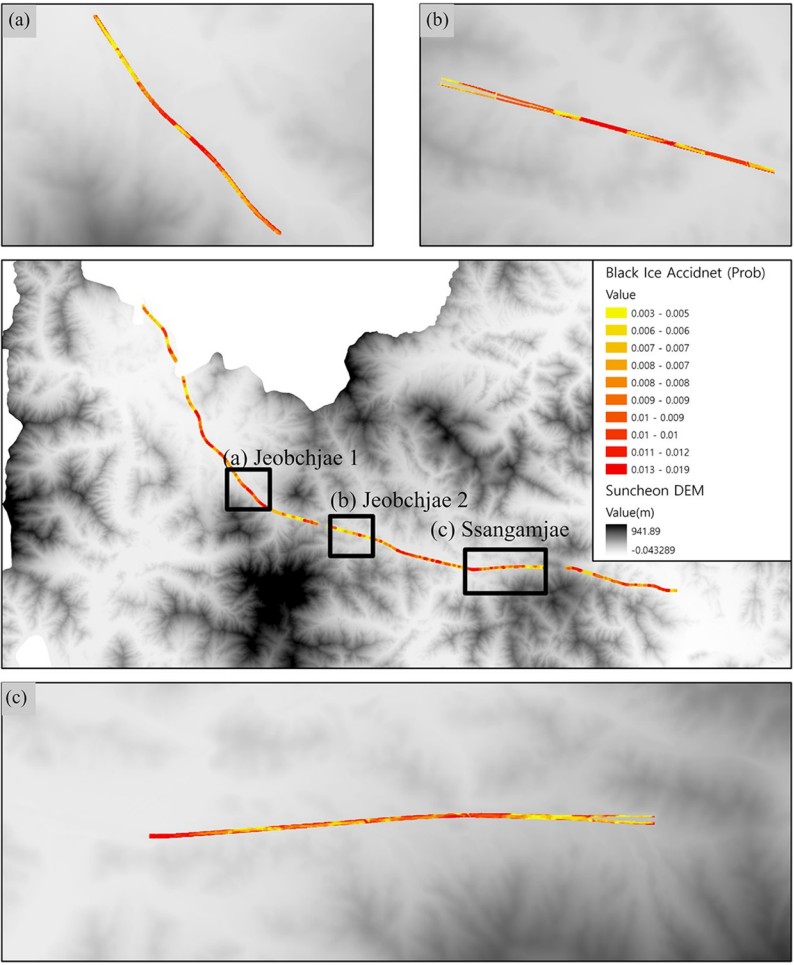

**Fig 8.** To conduct the hypothesis testing (Z-test), segments with a high frequency of accidents were selected: (a) black ice accident probability map for the Jeobchijae 1 section, (b) black ice accident probability map for the Jeobchijae 2 section, and (c) black ice accident probability map for Ssangamjae. (Source: DEM created by the LIDAR method at the National Geographic Information Institute in Korea, https://www.ngii.go.kr/kor/content.do?sq=204).

H0: The actual black ice vulnerability zones do not have a higher predicted accident probability than the general zones.

H1: The actual black ice vulnerable zones have a higher predicted accident probability than general zones.

By calculating the probability value of the test statistic using the p-value and setting the significance level to 0.05 with a 95% confidence level, it was determined that if the p-value was less than 0.05, the null hypothesis (H0) was rejected in favor of the alternative hypothesis (H1). In this study, we compared accident probabilities between the Jeollanam-do region and the general zones of the Honam Expressway. The accident probability in the general zone was predicted to be 0.0057. By comparing this with the accident probability in the Jeollanam-do region, we calculated a MAPE of 1.83% and a confidence level of 98.17%. Based on these findings, we selected the general zones of the Honam Expressway that demonstrated a high level of confidence as the population for hypothesis testing. Table 7 presents the p-values calculated from the Z-scores, which revealed that the P-values for Jeobchijae 1, Jeobchijae 2, and

**Table 7. Hypothesis testing (Z-test) results for the actual black ice accident-prone zones.**

| Z-test items | Jeobchijae 1 | Jeobchijae 2 | Ssangamjae |
|---|---|---|---|
| Sample mean (μ) | 0.006466535 | 0.006099772 | 0.005856457 |
| Sample siz (n) | 1502 | 272.00 | 1477.00 |
| Population mean (μ0) | 0.00573960040 | 0.00573960040 | 0.00573960040 |
| Population standard deviation | 0.001907 | 0.001907 | 0.001907 |
| Test statistic (Z) | 20.11 | 3.12 | 2.36 |
| P-value | 0.00 (<0.05) | 0.00092 (<0.05) | 0.01 (<0.05) |

Ssangamjae were < 0.05, thereby rejecting the null hypothesis (H0) and accepting the alternative hypothesis (H1). Consequently, the Z-test results indicate a higher accident probability for the black ice-prone zones of the Honam Expressway than for the general zones.

## 5.2. Discussion on countermeasures for black ice accident-inducing factors (Discussion ②)

An analysis of black ice accident-inducing factors is necessary to manage vulnerable zones with high accident probabilities. To address this issue, a sensitivity analysis was performed using Monte Carlo simulations. Table 8 presents the results of the study. The simulation data covered the winter of 2021 and were considered representative, whereby random variables were used to represent the distribution of each factor. The sensitivity was calculated by inputting random variables for each factor into the numerical model of system dynamics to measure the extent of the black ice variation. This was performed for each factor using a standard normal distribution while keeping the other factors fixed. The results show that the factor with the highest sensitivity was wind speed, with a value of 0.354, whereas the lowest sensitivity was observed for hill shade (0.085). Hill shade is known as a major triggering factor for black ice, but when the sensitivity analysis was performed after fixing the other major factors, such as the air temperature, to a standard normal distribution ($\mu = 0$, $\sigma = 1$), hill shade exhibited the lowest sensitivity. This indicates that while hill shade is a necessary condition for black ice occurrence in conjunction with other factors, its impact is minimal. Therefore, mitigating other major factors in low-hill-shade environments is considered more efficient than eliminating hill shades. The factors were ranked from 1 to 12, and the potential countermeasures for each factor were examined. The following were evaluated as potential countermeasures: early warning [15],

**Table 8. Sensitivity ranking and analysis of black ice accident factors.**

| Factors | Sensitivity | Rank | Measures | Effect | Type |
|---|---|---|---|---|---|
| Wind Speed | 0.354 | 1 | 'Early Warning | Reduction of vehicle velocity | Reduction of impact |
| Air Temp | 0.270 | 2 | Road Heating | Increase in road temperature | Reduction of impact |
| Angle | 0.203 | 3 | Sharp Curve Sign | Reduction of vehicle velocity | Reduction of impact |
| Slope | 0.200 | 4 | Slope sign | Reduction of vehicle velocity | Reduction of impact |
| Curvature | 0.198 | 5 | Curvature sign | Reduction of vehicle velocity | Reduction of impact |
| Vapor Pressure | 0.196 | 6 | 'Early Warning | Reduction of vehicle velocity | Reduction of impact |
| Velocity | 0.196 | 7 | Traffic velocity management | Reduction of vehicle velocity | Reduction of impact |
| Traffic | 0.196 | 7 | Route adjustment | Variance of traffic volume | Reduction of impact |
| Cloud Cover | 0.192 | 9 | 'Early Warning | Reduction of vehicle velocity | Reduction of impact |
| Precipitation | 0.160 | 10 | 'Early Warning | Reduction of vehicle velocity | Reduction of impact |
| Snow (Melted) | 0.140 | 11 | Road Heating | Ice formation prevention | Elimination of cause |
| Hillshade | 0.085 | 12 | Road Heating | Increase in road temperature | Elimination of cause |

road heating [44], sharp curve sign [45], slope sign [45], route adjustment [46], traffic velocity management [17, 47, 48], and curvature sign [45]. Among these, the measure with the highest frequency and proportion was early warning. This indicates that providing prior warnings to reduce vehicle speed and alert drivers is the most effective measure for mitigating the probability of black ice accidents at these road locations.

## 6. Conclusion

Monte Carlo simulations based on random variables are used in this study to predict the probability of black ice accidents occurring at different road locations. Furthermore, a sensitivity analysis, a key feature of Monte Carlo simulations, was employed to identify the factors contributing to black ice accident occurrences within the accident probability model. The analysis revealed that the average black ice value was 59.68 $g/m^2$ with a 15.25 $g/m^2$ standard deviation. The average black ice accident probability was 0.0058, with a standard deviation of 0.0019. The predicted probability of black ice accidents is the fundamental outcome of this study. Hypothesis testing using the Z-test was performed on vulnerable zones, namely Jeobchijae 1, Jeobchijae 2, and Ssangamjae. The p-values obtained were below the significance level of 0.05, thereby proving the validity of the model. Based on the validated model, a Monte Carlo simulation sensitivity analysis was conducted to identify the key factors influencing black ice accidents. Among these factors, wind speed exhibited the highest sensitivity, whereas hill shades exhibited the lowest. These findings can contribute to the formulation of effective strategies for mitigating black ice accidents in vulnerable areas, as identified using GIS.

This study integrated GIS and system dynamics for the prediction of black ice accidents and analyzed their contributing factors. Wind speed emerged as the most influential factor, and the primary solution proposed was the implementation of warnings. Although risk factors were identified in this study and discussions on potential solutions were initiated, further research is required to evaluate the effectiveness of the countermeasures in black ice accident-prone areas. Additionally, exploring advanced technologies, such as big data, artificial intelligence, and black ice multisensors, is essential for developing critical warning countermeasures. In the case of the hillshade factor, despite being a value that changed over time, the average value was input into the model, presenting a limitation in that the model was not designed as a time series. It is necessary to conduct a sensitivity analysis using time-series modeling techniques, such as system dynamics, and to compare the results with the sensitivity analysis of this study, which was conducted using the R language. If the model is designed as a time series, it is presumed that phenomena such as the thawing of black ice over time owing to solar radiation energy can also be simulated. The results of this study were derived by creating a numerical model capable of determining black ice conditions and then utilizing historical data related to black ice under the assumption that meteorological conditions would repeat in a similar fashion. This concept involves predicting or estimating future black ice occurrences, locations, and black ice accident probabilities using past data. However, for more accurate predictions, the model can be further developed into a time series, considering the applicability of "context history" and "context prediction." Context prediction is a technique that uses context history data to contextually forecast future situations. By utilizing various historical road-related data (air temperature, cloud cover, vapor pressure, precipitation, wind speed, snow, velocity, and traffic), we found that the accuracy of black ice occurrence predictions could be enhanced. Future research will be crucial for developing more sophisticated pattern analyses and predictive models using such contextual data. By analyzing various contextual elements over time, the likelihood of black ice formation at specific times and locations can be predicted more precisely. Examples include "risk management through the analysis of similarities in context

history [49], predictive models in a computing environment based on context history data [13, 14, 50], techniques for searching and monitoring sequential patterns in a context history database [51], and studies supporting the development of individual worker competencies based on context history [52]." Such approaches offer new strategies for road safety management related to black ice. Despite its limitations, this study offers valuable insights for government agencies (e.g., road traffic authorities) in managing accidents caused by black ice.

## Supporting information

**S1 File. Data table for the "Angle" factor on the Honam Expressway.** 10.5281/ zenodo.10863284.
(XLSX)

**S2 File. Data table for the "Bridge" factor on the Honam Expressway.** 10.5281/ zenodo.10863284.
(XLSX)

**S3 File. Data table for the "Curvature" factor on the Honam Expressway.** 10.5281/ zenodo.10863284.
(XLSX)

**S4 File. Data table for the "IC" factor on the Honam Expressway.** 10.5281/zenodo.10863284.
(XLSX)

**S5 File. Data table for the "Lake" factor on the Honam Expressway.** 10.5281/ zenodo.10863284.
(XLSX)

**S6 File. Data table for the "River System" factor on the Honam Expressway.** 10.5281/ zenodo.10863284.
(XLSX)

**S7 File. Data table for the "Slope" factor on the Honam Expressway.** 10.5281/ zenodo.10863284.
(XLSX)

## Acknowledgments

Special thanks to Professor Hong Sik Yun of Sungkyunkwan University, the corresponding author and the mentor of the knowledge for this paper.

## Author Contributions

**Conceptualization:** Seok Bum Hong.

**Data curation:** Seok Bum Hong.

**Formal analysis:** Hong Sik Yun.

**Funding acquisition:** Hong Sik Yun.

**Investigation:** Seok Bum Hong.

**Methodology:** Hong Sik Yun.

**Project administration:** Seok Bum Hong.

**Resources:** Hong Sik Yun.

**Software:** Seok Bum Hong.

**Supervision:** Hong Sik Yun.

**Validation:** Hong Sik Yun.

**Visualization:** Seok Bum Hong.

**Writing – original draft:** Seok Bum Hong.

**Writing – review & editing:** Hong Sik Yun.

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
