## [Decision Letter · Decision Letter 0]

28 Dec 2023

PONE-D-23-32679Predicting Black Ice-Related Accidents with Probabilistic Modeling Using GIS-Based Monte Carlo SimulationPLOS ONE

Dear Dr. Yoon,

Thank you for submitting your manuscript to PLOS ONE. After careful consideration, we feel that it has merit but does not fully meet PLOS ONE’s publication criteria as it currently stands. Therefore, we invite you to submit a revised version of the manuscript that addresses the points raised during the review process.

We look forward to receiving your revised manuscript.

Kind regards,

Abel C.H. Chen

Academic Editor

PLOS ONE

“This research was also supported by the National Research Foundation of Korea (NRF) grant funded by the Korea government (MSIT) (No. 2021R1A2C201231911).”

4. PLOS requires an ORCID iD for the corresponding author in Editorial Manager on papers submitted after December 6th, 2016. Please ensure that you have an ORCID iD and that it is validated in Editorial Manager. To do this, go to ‘Update my Information’ (in the upper left-hand corner of the main menu), and click on the Fetch/Validate link next to the ORCID field. This will take you to the ORCID site and allow you to create a new iD or authenticate a pre-existing iD in Editorial Manager. Please see the following video for instructions on linking an ORCID iD to your Editorial Manager account: https://www.youtube.com/watch?v=_xcclfuvtxQ.

5. We note that Figures 3, 6 and 8 in your submission contain [map/satellite] images which may be copyrighted. All PLOS content is published under the Creative Commons Attribution License (CC BY 4.0), which means that the manuscript, images, and Supporting Information files will be freely available online, and any third party is permitted to access, download, copy, distribute, and use these materials in any way, even commercially, with proper attribution. For these reasons, we cannot publish previously copyrighted maps or satellite images created using proprietary data, such as Google software (Google Maps, Street View, and Earth). For more information, see our copyright guidelines: http://journals.plos.org/plosone/s/licenses-and-copyright.

1. You may seek permission from the original copyright holder of Figures 3, 6 and 8 to publish the content specifically under the CC BY 4.0 license. 

Reviewers' comments:

Reviewer's Responses to Questions

**Comments to the Author**

1. Is the manuscript technically sound, and do the data support the conclusions?

Reviewer #1: Yes

2. Has the statistical analysis been performed appropriately and rigorously? 

Reviewer #1: Yes

3. Have the authors made all data underlying the findings in their manuscript fully available?

Reviewer #1: Yes

4. Is the manuscript presented in an intelligible fashion and written in standard English?

Reviewer #1: Yes

5. Review Comments to the Author

Reviewer #1: The theme approached in the paper is relevant. The main strong points of the work are: (1) the theme is current and strategic involving aspects related to accident prevention and strategies to prediction; (2) the general quality of the text and more specifically the quality of the results (section 3) and their discussion (section 4). The main weak points are: (1) the little discussion of the scientific contribution of the work (comparison with state of the art); (2) the discussion of related works was restricted to describe the researches without a consistent comparison and discussion of the scientific contribution of this paper. The discussion is restricted to last paragraphs of Section 1 but without a more detailed comparison with related works. Table 1 compares the proposal with only one work.

During the article revision, I found aspects that must be considered in a future version. I will list them in the same order they appeared during the reading. I hope the comments can be used to improve the text. At the end, I will present my final evaluation about the work.

1) The abstract must be improved to clarify the scientific contribution of the paper. The authors discuss several aspects of the work, but what is the difference in relation to related works (state-of-the-art)? The reader ends the abstract without know what is the scientific contribution of the proposal;

2) It is important to include in the section "Introduction", a more complete description of the paper’s proposal. The section discusses several topics, but one of the most important aspects of a scientific paper is not presented, namely, the scientific contribution based on a comparison with related works. There is a discussion of related works and a table with only one work produced also in South Korea, but I recommend improving the discussion, mainly amplifying the table. In addition, the authors can include a “research question” to indicate the focus of the research. Based on the question, the reader can understand the contribution that is being sought;

3) I believe the following aspect is the most relevant weakness of the paper. Section 1 does not sufficiently compare the related works with the proposal. There is an initial discussion, but I believe that it would be interesting to improve the organization through a “comparison with the proposed work”. Typically, based on a revision of related works is possible to indicate the contribution sought by an article. I advise the authors to improve the comparison table (table 1) with more works. Based on this table, it would be easier to understand the relation between the works, their relationship with the proposal and the scientific contribution of this research. Perhaps, this discussion can be included in a specific Section 2 dedicated to “Related Works”;

4) I recommend the authors to enrich the article through the inclusion of a discussion and references related to “Contexts” and more specifically about two emergent research topics called “Context Histories” and “Context Prediction”. Context histories are time series of Contexts that can be used to several kinds of data analysis such as Context Prediction and Similarity Analysis. The authors can explore these topics as future works applying these strategies on their work, mainly for recording in context histories the history information related to the roads (for example, air temperature, precipitation, snow, traffic) and using these context histories to several kind of analysis, including “Context Prediction” and “Pattern Analysis” (considering conditions that can generated Black Ice). Section 5 can be improved with future works regarding these strategic themes. I recommend using the following references to indicate this research way. These references involve the definition of context histories, context prediction, similarity analysis of context histories and pattern discovery on context histories: (1) A Multi-Temporal Context-aware System for Competences Management. International Journal of Artificial Intelligence in Education, 2015. https://doi.org/10.1007/s40593-015-0047-y; (2) ORACON: An adaptive model for context prediction. Expert Systems with Applications, 2016. https://doi.org/10.1016/j.eswa.2015.09.016; (3) A Risk Prediction Model for Software Project Management based on Similarity Analysis of Context Histories. Information and Software Technology, 2021. https://doi.org/10.1016/j.infsof.2020.106497; (4) CHSPAM: a multi-domain model for sequential pattern discovery and monitoring in contexts histories. Pattern Analysis and Applications, 2020. https://doi.org/10.1007/s10044-019-00829-9.

5) Section 2 describes the methodology. The first aspect that must be answered is why the Monte Carlo simulation was chosen to make the predictions in this proposal. There are several other strategies to conduct predictions. In addition, it is important to explain how GIS was used in the methodology and why it was used. Moreover, Figure 2 and Table 2 must be better explained. In Section 2.2.1, the authors must clarify how the actual data indicated were acquired and organized. Table 4 has a general vision, but I recommend to better explain this step of the research. Can that kind of information be treated as Context information (see the item 4 in this review)?

6) Section 3 presents the results. I recommend in this section to better describe why the results are relevant to the scientific contribution of this research. In addition, the text indicates three results, but it presents only two (subsections 3.1 and 3.2);

7) In section 4, firstly, I recommend including an introduction (between titles 4 and 4.1). I believe that section can be improved with an initial discussion about the relation of the results with the scientific contribution and with the research question if this question was included in the “Introduction” as recommend in this review. I also recommend summarizing the main results to simplify the reading and better organize them. For example, the authors can include at the end of the section a table dedicated to present the main lessons learned in the work. I also recommend including a discussion of difficulties encountered and limitations of the research;

8) Finally, I advise the authors to improve the "Conclusions" (Section 5) through a more complete discussion of future works. As indicated in the item 4 of this review, the authors can propose future work based on Context and analysis of Context histories to predict contexts related to Black Ice. They already have indicated in the text the interest in time-series modeling techniques. Context histories are time-series that organized information related to Contexts. These themes are aligned with the research presented in this article.

Based on these comments, I can present my final evaluation. I think the article can be accepted, after the improvements indicated in this review. I consider that that paper will need “Major Revision”.

6. PLOS authors have the option to publish the peer review history of their article (what does this mean?). If published, this will include your full peer review and any attached files.

Reviewer #1: No

---

## [Author Response · Author response to Decision Letter 0]

1 Apr 2024

Manuscript ID: PONE-D-23-32679

My co-authors and I would like to express our gratitude to the reviewers for their constructive feedback and suggestions for strengthening our research. The changes we have made to the attached file in response to such feedback and suggestions have been highlighted in blue to facilitate their identification. I would also like to offer my apologies for the length of time it took us to prepare this response.

Journal requirements

Our response: Thank you for your detailed review, Editor. I have accessed the links provided to ensure that my manuscript meets the style requirements. Since there are only two authors in my paper, I did not include a symbol to denote equal contribution in the author byline. Additionally, there are no Consortia or other Group Authors involved, so that has not been indicated either. The current information about the authors in my manuscript is as follows.

Seok Bum Hong 1, Hong Sik Yun 12*

1 Interdisciplinary Program for Crisis, Disaster and Risk Management, Sungkyunkwan University, Suwon, Gyeonggi Province, Republic of Korea

2 School of Civil and Architectural Engineering, Sungkyunkwan University, Suwon, Gyeonggi Province, Republic of Korea

* Corresponding author

Email: yoonhs@skku.edu

Our response: Thank you for your meticulous review, Editor. I have reviewed the guidelines to ensure that the code I have generated is made available without restrictions upon publication. My code (Black Ice Accidents Prediction_R code) will be shared in accordance with best practices, facilitating reproducibility and reuse, through the Google Drive link listed in the Data Availability Statement within the manuscript.

(line 530) Data Availability Statement

The Black Ice Prediction code used in this study is available without restrictions at the following link: https://zenodo.org/records/10725620?token=eyJhbGciOiJIUzUxMiJ9.eyJpZCI6IjA5ZTAyYWM4LWQ1NTktNDgyOC05ZTNhLWQxMWZiNTU4NmZkNCIsImRhdGEiOnt9LCJyYW5kb20iOiJjMjM0NTg3Y2ZjYmVjZTJmZDEzZDc5NDI5Y2YxNWUzMSJ9.erX0WRZiuKi_qlxhk66UCD7jaGdFkdX3PoRP4ZF12oCkxzhKIetGw51X9nOIg3GxtYkzCqhuOS9hY9IbzuRMXQ

“This research was also supported by the National Research Foundation of Korea (NRF) grant funded by the Korea government (MSIT) (No. 2021R1A2C201231911).”

Our response: Thank you for your correction, Editor. We would like our funding statement to be amended as follows. Additionally, having this statement in the Acknowledgments section of the paper would facilitate our funding process. Should our paper be successfully published, could you advise on how we might officially verify the inclusion of the following text? Is there a possibility to at least make the project number (No. 2021R1A2C201231911) visible in the Acknowledgments of the paper?

“This research was also supported by the National Research Foundation of Korea (NRF) grant funded by the Korea government (MSIT) (No. 2021R1A2C201231911).”

Would it be permissible, for instance, to write the following in the Acknowledgments section?

“We gratefully acknowledge the support of the National Research Foundation of Korea (No. 2021R1A2C201231911).”

4. PLOS requires an ORCID iD for the corresponding author in Editorial Manager on papers submitted after December 6th, 2016. Please ensure that you have an ORCID iD and that it is validated in Editorial Manager. To do this, go to ‘Update my Information’ (in the upper left-hand corner of the main menu), and click on the Fetch/Validate link next to the ORCID field. This will take you to the ORCID site and allow you to create a new iD or authenticate a pre-existing iD in Editorial Manager. Please see the following video for instructions on linking an ORCID iD to your Editorial Manager account: https://www.youtube.com/watch?v=_xcclfuvtxQ.

Our response: Thank you for your guidance, Editor. The corresponding author, Dr. Hong Sik Yun, has updated the ORCID (0000-0002-2104-9423) in the 'Update My Information' section within Editorial Manager as instructed. 

5. We note that Figures 3, 6 and 8 in your submission contain [map/satellite] images which may be copyrighted. All PLOS content is published under the Creative Commons Attribution License (CC BY 4.0), which means that the manuscript, images, and Supporting Information files will be freely available online, and any third party is permitted to access, download, copy, distribute, and use these materials in any way, even commercially, with proper attribution. For these reasons, we cannot publish previously copyrighted maps or satellite images created using proprietary data, such as Google software (Google Maps, Street View, and Earth). For more information, see our copyright guidelines: http://journals.plos.org/plosone/s/licenses-and-copyright.

Our response: Thank you very much for your meticulous review. The base map (World Imagery (WGS84)) from ArcGIS that we initially utilized is subject to a license that adheres to the conditions for the use of items owned by Esri and therefore does not comply with the CC BY 4.0 license. Consequently, we have transitioned the background to a visualization map based on Digital Elevation Models (DEM). The DEM data, which is permitted for reprocessing, was obtained from the National Geographic Information Institute in Korea. We assessed that replacing the original aerial maps—not only because they do not meet the licensing requirements but also because a shift to more meaningful information could enhance the representation of results in our paper. Therefore, Figure 3 has been updated to feature a Hillshade based on DEM, and Figures 6 and 8 have been revised to DEM, with numerical values provided in the legends. The revised figures are as follows. Additionally, we have disclosed the source of the DEM data used to construct the background in the captions of these figures.

Fig. 3. Honam Highway and Spatial Information in Suncheon, Jeollanam-do. (Source: DEM created by the LIDAR method at the National Geographic Information Institute in Korea, https://www.ngii.go.kr/kor/content.do?sq=204)

Fig. 6. Estimation map of black ice accident probability. (Source: DEM created by the LIDAR method at the National Geographic Information Institute in Korea, https://www.ngii.go.kr/kor/content.do?sq=204)

Fig. 8. To conduct the hypothesis testing (Z-test), high-frequency accident segments were selected, namely, (a) the black ice accident probability map for the Jeobchijae 1 section, (b) the black ice accident probability map for the Jeobchijae 2 section, and (c) the black ice accident probability map for Ssangamjae. (Source: DEM created by the LIDAR method at the National Geographic Information Institute in Korea, https://www.ngii.go.kr/kor/content.do?sq=204)

Reviewer #1

The theme approached in the paper is relevant. The main strong points of the work are: (1) the theme is current and strategic involving aspects related to accident prevention and strategies to prediction; (2) the general quality of the text and more specifically the quality of the results (section 3) and their discussion (section 4). The main weak points are: (1) the little discussion of the scientific contribution of the work (comparison with state of the art); (2) the discussion of related works was restricted to describe the researches without a consistent comparison and discussion of the scientific contribution of this paper. The discussion is restricted to last paragraphs of Section 1 but without a more detailed comparison with related works. Table 1 compares the proposal with only one work. During the article revision, I found aspects that must be considered in a future version. I will list them in the same order they appeared during the reading. I hope the comments can be used to improve the text. At the end, I will present my final evaluation about the work.

1. The abstract must be improved to clarify the scientific contribution of the paper. The authors discuss several aspects of the work, but what is the difference in relation to related works (state-of-the-art)? The reader ends the abstract without know what is the scientific contribution of the proposal;

Our response: Thanks for your insightful comment. In the introduction, we have added recent research trends, and in the abstract, we have further emphasized the scientific contribution. The added part in the abstract is as follows.

(line 26) The scientific contribution of this study lies in the development of a method beyond simple road temperature predictions for evaluating the risk of black ice occurrences and subsequent accidents. By employing Monte Carlo simulations, the probability of black ice accidents can be predicted more accurately through decoupling meteorological and traffic factors over time. 

2. It is important to include in the section "Introduction", a more complete description of the paper’s proposal. The section discusses several topics, but one of the most important aspects of a scientific paper is not presented, namely, the scientific contribution based on a comparison with related works. There is a discussion of related works and a table with only one work produced also in South Korea, but I recommend improving the discussion, mainly amplifying the table. In addition, the authors can include a “research question” to indicate the focus of the research. Based on the question, the reader can understand the contribution that is being sought;

Our response: Thank you for your good comment. As per the suggestions of the reviewer, we have enhanced the comparison with related works in the section 2. Furthermore, in Table 1, we have included studies categorized into various areas such as Road temperature prediction, black ice index prediction, and black ice prediction. Lastly, we have incorporated research questions that elucidate the focus of our study. 

The details added to the paper concerning the comparison with related works are as follows:

(line 86) Previous studies proposed low-cost sensors using electrical conductivity and GIS visualization technologies based on sensor detection for black ice detection and management[15]. This study has the capability of monitoring the black ice conditions of roads in real time based on sensors; however, because long-term predictions are not provided, there is a time limitation for devising effective countermeasures. Numerical weather prediction models have been used to predict road surfaces and traffic conditions[16]. This study provides an approximate prediction of the road surface conditions but faces limitations owing to inadequate integration with GIS and challenges in accurately calculating the number of occurrences. Previous studies created models of vehicle speed distributions suitable for various weather and traffic conditions on highways, thereby identifying more dangerous weather conditions[17]. In this study, the hazardous conditions for the occurrence of black ice were numerically predicted; however, the method had limitations in predicting the occurrences and locations of black ice based on GIS. A block diagram model has also been developed to estimate the ice index using road surface temperature, air temperature, and humidity[18-20]. These studies predicted the ice index to be within an approximate range; however, limitations existed in forecasting the occurrence and location of black ice based on GIS. These studies commonly lack methods for predicting the occurrence of black ice and the accident itself, and identifying the factors triggering black ice accidents is insufficient. Furthermore, even with predictions for the occurrence of black ice, the analysis often remains limited to the road temperature or the black ice index, and there is an issues with representing the occurrence and location of black ice in a GIS, which is linked with spatial information.

(line 118) Monte Carlo simulations generate various scenarios by considering the input parameters as probability distributions independent of time and using the average of the outcomes derived from these scenarios as predictions for disaster occurrence. In relation to the Monte Carlo method applied in the field of disasters, there have been studies predicting traffic accidents using statistical input values[25, 26], and studies predicting meteorological disasters such as tornadoes and floods using statistical input values [10, 11]. These studies have provided directions for the application of Monte Carlo simulations in disaster management. However, research related to traffic accidents has not been effectively integrated with GIS and has not primarily focused on black ice accident cases. Considering prior research, it was deemed necessary to apply Monte Carlo simulations to the field of black ice accident prediction, particularly by utilizing meteorological and traffic factors as key inputs. 

The content added to the paper regarding the research questions that represent the focus of the study is as follows:

(line 126) Previous research has raised the question of which theories should be used to predict and respond to traffic accidents caused by black ice. This emphasizes the need for a GIS-based analysis of black ice occurrences and locations, numerical modeling for black ice accident probability, and the use of Monte Carlo simulations for time independence.

The improvements made to the discussion by expanding Table 1 are as follows: 

(line 130) Table 1 lists the improvements of this study compared with recent research. As shown in Table 1, the scientific contribution of this study has several respects: the introduction of a GIS in the field of road temperature prediction, the enhancement of the simple black ice index by predicting the occurrences and locations of black ice, the prediction of the probability of black ice accidents beyond mere forecasts of black ice occurrences, and the conduction of time-independent simulations (Monte Carlo simulation) by inputting traffic and meteorological factors with probabilistic distributions owing to their uncertainty.

Table 1. Developments of this study compared with previous studies.

 Study Area Factors 

(added) Scenario Simulation Results

Previous 

Study ① 

[27]

Jerash city, Jordan - Time-Dependent

(Specific Dates) Numerical 

Model Road temperature

(No GIS)

Previous 

Study ②

[16] 

Finland - Time-Dependent

(Specific Period) Numerical 

Model Road temperature

(No GIS)

Previous 

Study ③ 

[20]

Gunsan, Jeollabuk-do, South Korea - Time-Dependent

(Specific Period) Flow chart Black ice index

(No GIS)

Previous 

Study ④ 

[14]

Suncheon, Jeollanam-do, South Korea - Time-Dependent

(Specific Date) Numerical 

Model Black Ice

(GIS)

P

---

## [Decision Letter · Decision Letter 1]

29 Apr 2024

Predicting Black Ice-Related Accidents with Probabilistic Modeling Using GIS-Based Monte Carlo Simulation

PONE-D-23-32679R1

Dear Dr. Yoon,

We’re pleased to inform you that your manuscript has been judged scientifically suitable for publication and will be formally accepted for publication once it meets all outstanding technical requirements.

Kind regards,

Abel C.H. Chen

Academic Editor

PLOS ONE

Additional Editor Comments (optional):

Reviewers' comments:

Reviewer's Responses to Questions

**Comments to the Author**

1. If the authors have adequately addressed your comments raised in a previous round of review and you feel that this manuscript is now acceptable for publication, you may indicate that here to bypass the “Comments to the Author” section, enter your conflict of interest statement in the “Confidential to Editor” section, and submit your "Accept" recommendation.

Reviewer #1: All comments have been addressed

2. Is the manuscript technically sound, and do the data support the conclusions?

Reviewer #1: Yes

3. Has the statistical analysis been performed appropriately and rigorously? 

Reviewer #1: I Don't Know

4. Have the authors made all data underlying the findings in their manuscript fully available?

Reviewer #1: Yes

5. Is the manuscript presented in an intelligible fashion and written in standard English?

Reviewer #1: Yes

6. Review Comments to the Author

Reviewer #1: I revised the article and the review letter.

The text has been significantly improved, mainly through new references and a better discussion of the scientific contribution.

7. PLOS authors have the option to publish the peer review history of their article (what does this mean?). If published, this will include your full peer review and any attached files.

Reviewer #1: No

---

## [Editor Report · Acceptance letter]

13 May 2024

PONE-D-23-32679R1 

PLOS ONE

Dear Dr. Yun, 

I'm pleased to inform you that your manuscript has been deemed suitable for publication in PLOS ONE. Congratulations! Your manuscript is now being handed over to our production team.

Kind regards, 

on behalf of

Dr. Abel C.H. Chen 

Academic Editor

PLOS ONE